# Development of a high-productivity, halophilic, thermotolerant microalga *Picochlorum renovo*

Lukas R. Dahlin[1], Alida T. Gerritsen[2], Calvin A. Henard [3], Stefanie Van Wychen[3], Jeffrey G. Linger[3], Yuliya Kunde[4], Blake T. Hovde[4], Shawn R. Starkenburg [4], Matthew C. Posewitz[1] & Michael T. Guarnieri[3]*

Microalgae are promising biocatalysts for applications in sustainable fuel, food, and chemical production. Here, we describe culture collection screening, down-selection, and development of a high-productivity, halophilic, thermotolerant microalga, *Picochlorum renovo*. This microalga displays a rapid growth rate and high diel biomass productivity ($34\,g\,m^{-2}\,day^{-1}$), with a composition well-suited for downstream processing. *P. renovo* exhibits broad salinity tolerance (growth at $107.5\,g\,L^{-1}$ salinity) and thermotolerance (growth up to $40\,°C$), beneficial traits for outdoor cultivation. We report complete genome sequencing and analysis, and genetic tool development suitable for expression of transgenes inserted into the nuclear or chloroplast genomes. We further evaluate mechanisms of halotolerance via comparative transcriptomics, identifying novel genes differentially regulated in response to high salinity cultivation. These findings will enable basic science inquiries into control mechanisms governing *Picochlorum* biology and lay the foundation for development of a microalga with industrially relevant traits as a model photobiology platform.

[1] Department of Chemistry, Colorado School of Mines, Golden, CO 80401, USA. [2] Computational Science Center, National Renewable Energy Laboratory, Golden, CO 80401, USA. [3] National Bioenergy Center, National Renewable Energy Laboratory, Golden, CO 80401, USA. [4] Los Alamos National Laboratory, Los Alamos, NM 87545, USA. *email: michael.guarnieri@nrel.gov

Microalgae are a source of renewable biomass and promising photosynthetic biocatalysts for the sustainable production of fuel and chemical intermediates[1]. Importantly, they are also valuable model systems for fundamental investigation of mechanistic photobiology[2]. These microbes possess a series of unique characteristics that make them well-suited for biotechnological applications, including year-round cultivation capacity in saline water on non-arable land, higher potential biofuel yields than terrestrial crops, and the ability to utilize $CO_2$ as a sole carbon source[3,4]. Rising greenhouse gas emissions from anthropogenic sources has led to a resurgent interest in exploiting these organisms for concurrent $CO_2$ capture and renewable biocommodity production[5,6]. However, at present, current model microalgal systems are not suitable for outdoor deployment, displaying low productivity under relevant environmental conditions (e.g., high light intensity, high temperature, seawater environments)[7]. Further, top-candidate deployment systems display low genetic throughput, often requiring weeks-to-months to generate and verify transgenic lines, which hinders fundamental mechanistic inquiry and metabolic engineering strategies in deployment-relevant microalgae[7,8].

Since its first classification in 2004, the genus *Picochlorum* has been recognized for its distinct characteristics of broad thermotolerance, salinity tolerance, compact genome architecture, fast doubling time, and resilience to high light intensity[6,9–13]. An alga of the genus *Picochlorum* was recently shown to have the highest biomass productivity in a comparative analysis between a series of industrially relevant microalgae, underscoring this genera's deployment potential[6]. However, to date, there are limited insights into *Picochlorum* halotolerance, biosynthetic capacity, biomass characterization, and genetic tractability, hindering its development as a fundamental platform and for biotechnological applications.

Here, we report the down-selection, characterization, and development of a novel alga of the genus *Picochlorum, Picochlorum renovo sp. nov.* We characterized the diel biomass productivity ($34\,g\,m^{-2}\,day^{-1}$) of this alga under simulated outdoor cultivation conditions, quantifying the protein, carbohydrate, and lipid content (20, 60, and 10% ash-free dry cell weight, respectively), thermotolerance (growth capacity up to 40 °C), and salinity tolerance (growth at $107.5\,g\,L^{-1}$ salinity). Furthermore, we generated nuclear, chloroplast, and mitochondrial genome sequences and report comparative transcriptomic analyses under low- and high-salt conditions, enabling high-resolution genome annotation and providing novel insight into the mechanisms of halotolerance. Lastly, we developed a set of facile genetic tools that enable expression of multiple transgenes inserted into either the nuclear or chloroplast genomes. Combined, these data will enable fundamental insights into *Picochlorum* photobiology and inform targeted genetic engineering strategies to accelerate microalgal biotechnological applications in a deployment-relevant, emerging model microalga.

## Results

### Down-selection, physiology, and compositional analysis.
We screened a >300-strain microalgal culture collection, in order to identify halotolerant strains[14,15]. Over 100 unique halotolerant isolates were down-selected and screened under simulated (diurnal light and temperature cycling) summer growth conditions using a custom built photobioreactor, described in Dahlin et al.[15]. We identified one isolate that exhibited a noticeably faster growth rate relative to other isolates, including control strains *Nannochloropsis oceanica* (KA32) and *Nannochloropsis salina* (CCMP 1776), two top-candidate strains currently under evaluation for outdoor deployment (Fig. 1)[15–17]. This rapidly

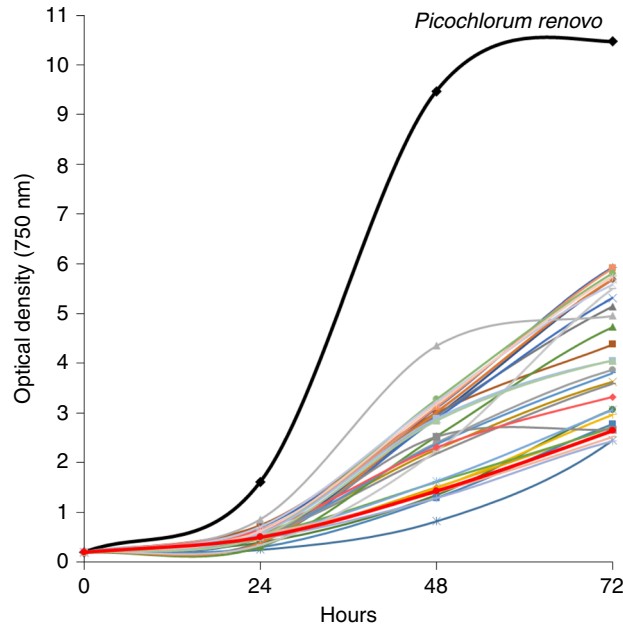

**Fig. 1** Representative culture collection growth screening data. The rapid growth and high optical density phenotype of *P. renovo* is highlighted in black. *Nannochloropsis salina* CCMP 1776 is bolded in red for reference

growing strain was down-selected for further analysis and development. Under batch growth this microalga displayed a diel biomass productivity of $34.3\,g\,m^{-2}\,day^{-1}$, from hour 6 to 30, representative of one day and night of high-productivity growth (Fig. 2a). Dark period biomass loss was $0.25\,g\,m^{-2}\,h^{-1}$ during the first 11-h dark period and $0.46\,g\,m^{-2}\,h^{-1}$ in the second. Cell division occurs during both light and dark periods when grown under a diel cycle. Cessation of cell division and biomass accumulation occurs simultaneously. Nitrogen supplementation during stationary phase led to growth reinitiation (Supplementary Fig. 1a). We observed peak growth at 35 °C under continuous illumination, with growth capacity up to 40 °C (Supplementary Fig. 1b).

Biomass composition varies as a function of growth phase, with fluctuations in carbohydrate and protein content observed throughout diel cycles (Fig. 2b). Notably there is a substantial decrease in glucose (derived from biomass hydrolysis) following inoculation into fresh media, with glucose declining from 52 to 1.4% of AFDW (ash-free dry weight) (Fig. 2c). Lipid content, as measured by fatty acid methyl esters, varied from 8.5 to 16.2%, with C16:0, C16:3, C18:1n9, C18:2n6, and C18:3n3 representing major lipid fractions (Fig. 2d). Thirty hours post-inoculation the cells enter stationary phase and have an ash-free composition of 10% FAME (fatty acid methyl ester), 20% protein, 59.5% carbohydrates (measured as hydrolyzed monomeric sugars), and 10.5% unidentified biomass components (Fig. 2b).

### Genomic analysis and speciation.
We conducted phylogenetic analysis of the isolate's 18S rRNA, identifying high similarity (>99%) to numerous *Picochlorum* species, providing initial evidence for taxonomic classification. PacBio genome sequencing generated an assembled nuclear genome containing 29 contigs with 14.4 Mbps and 46.2% GC, similar to previously reported *Picochlorum* genomes[12,13]. In total, 8902 protein coding sequences were putatively annotated, with an average of 2.2 exons/1.2 introns per gene. The nuclear genome contains a homolog of the universally conserved meiosis associated gene,

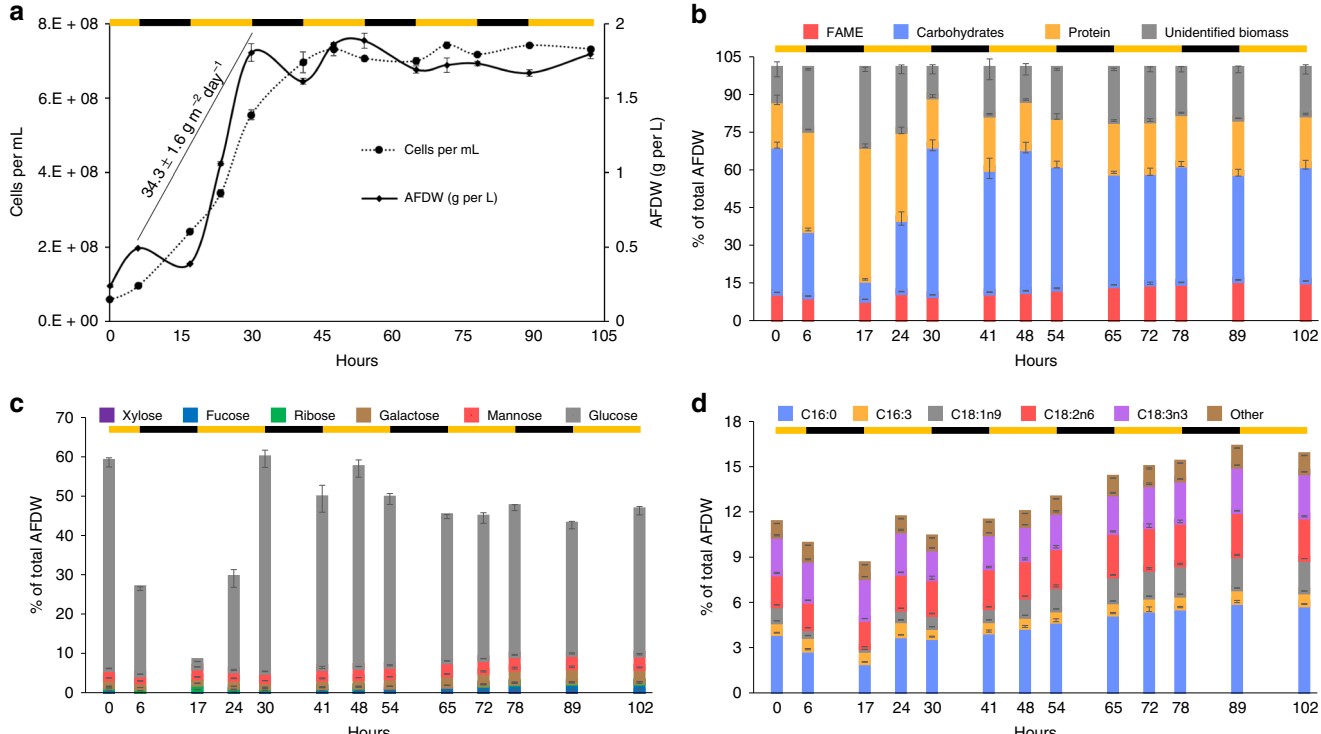

**Fig. 2** Overview of *P. renovo* productivity and associated biomass composition, as a function of time. Alternating black and yellow bars depict the light-dark growth cycle. **a** Growth curves as a function of ash-free dry weight (g per L) and cell density (cells per mL). Areal productivity is shown for hour 6 to 30, representing one light-dark cycle (day). **b** The biomass content of lipid (as FAMEs), carbohydrate, protein, and the fraction of biomass not identified. **c** Carbohydrate speciation via hydrolysis of biomass. **d** Fatty acid speciation via fatty acid methyl ester analysis, representative of the lipid fraction of the biomass. All data points are an average of $n = 3$ biological replicates; error bars depict the standard deviation of the replicates

*spo11–2* (E-value: 2E-22), and homologs of multiple meiotic and/ or homologous recombination associated genes (with reported E-values), including four *rad51* homologs (2E-107 to 4E-26), *dmc1* (2E-121), *pol2A* (0), *rfc1* (4E-34), *polD1* (0), *mre11* (2E-77), *rad50* (0), *rad54* (4E-134), *mus81* (6E-15), *msh4* (2E-10), *msh5* (2E-50), *rpa1* (3E-69), *rpa2* (8E-25), and *rpa3* (5E-16)[18]. Further evidence of genes putatively involved in meiosis is provided by the identification of *oda2* (0) and *bug22* (1E-72) homologs, which are flagella associated genes, implicated in gamete pairing prior to mating[19–23]. We also identified a putative chlorophyllide-a oxygenase (0), necessary for chlorophyll b production, and cell division was observed to occur by autosporulation (Supplementary Fig. 2), providing additional evidence for classification of this strain as a *Picochlorum*[11]. When compared to other available *Picochlorum* genomes, the novel *Picochlorum* isolate displayed 87–94% whole genome sequence identity (Supplementary Table 1). These data support that the isolate is a novel *Picochlorum* species, henceforth termed *Picochlorum renovo sp. nov.*[24].

Chloroplast and mitochondria genome maps are presented in Supplementary Fig. 3. The 74 kb chloroplast genome displayed a non-canonical chloroplast genome architecture lacking an inverted repeat region, as noted by Krasovec et al. for the genus *Picochlorum*[12]. The 36 kb mitochondrial genome displayed a compact coding architecture, representing the highest mitochondrial coding density reported to date (1.05 CDS per kb) for the class *Trebouxiophyceae*, in line with the previously reported mitochondrial genome of *Picochlorum costavermella*[12]. Notably, genes encoding a homing endonuclease and protein of unknown function split the mitochondrial 23s rRNA, contributing, in part, to the higher coding density (Supplementary Fig. 3).

**Transcriptome response to salinity**. We observed broad halotolerance in *P. renovo*, as reported previously for this genus[10,11], with cultivation capacity in minimal media salinity concentrations ranging from 8.75 to 107.5 g $L^{-1}$ sea salts (Supplementary Fig. 1c, d). To better understand the genes involved in the salinity response, cultures were grown in 8.75 and 35 g $L^{-1}$ salinity seawater and assessed via comparative transcriptomics. RNA from triplicate mid-log phase cultures was sequenced, and subsequent differential expression analysis was performed. In total, 3464 genes were differentially expressed at 35 g $L^{-1}$ salinity (1934 down, 1530 up) at statistically significant values ($q < 0.05$), representing 39 percent of total coding sequences (Supplementary Data File 1). Gene ontology semantic analyses were used to deconvolute the large number of differentially expressed transcripts, implicating a subset of processes involved in the high-salt response, including previously reported genes governing proline metabolism (Supplementary Fig. 4)[10]. A series of previously unreported haloresponsive genes were also observed, including *ppsA* (E-value: 2E-54), *ppsC* (E-value: 2E-60), *pks1* (E-value: 4E-30) and *pks15* (E-value: 1E-28) (polyketide synthases), *iput1* (inositol phosphorylceramide glucuronosyltransferase, E-value: 9E-75), *cerk* (ceramide kinase, E-value: 8E-34), *rad54* (DNA repair and recombination protein, E-value: 2E-20), and *dmc1* (disrupted meiotic cDNA 1, E-value 2E-121), discussed further below.

**Nuclear and chloroplast engineering**. We randomly integrated a linear PCR amplicon comprised of native promoter and terminator elements into the nuclear genome of *P. renovo*, directing transcription of 2A peptide-linked bleomycin resistance gene and the fluorescent reporter *mcherry* (Fig. 3a) via electroporation. mCherry was chosen as a reporter gene for nuclear expression

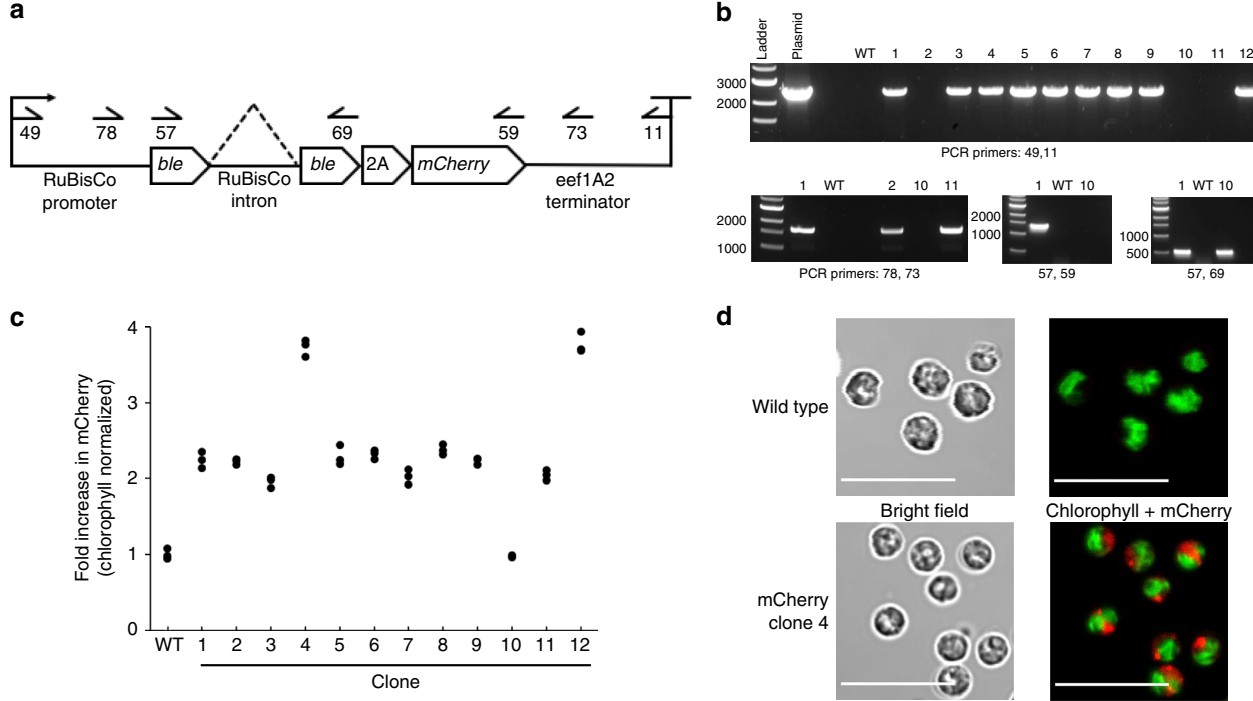

**Fig. 3 Overview of *P. renovo* nuclear transformation. a** Construct design showing genetic elements and primers used to generate DNA for electroporation (49 and 11) and subsequent PCR confirmation of transformants. **b** PCR verification of 12 clones utilizing primers shown in panel **a**. **c** Dot plot of fluorescent plate reader data of wild type and mCherry transformants, normalized to chlorophyll autofluorescence. Data is from three biological replicates. **d** Confocal microscopy images of wild type and transformant microalgae expressing mCherry. Green coloring represents chlorophyll autofluorescence, red coloring represents mCherry fluorescence, 10 μm scale bar

based on prior reports of high signal to noise ratios in microalgae[25].

Per transformation, an average of 41 colonies were obtained, representing transformation efficiencies of 14 colonies per μg of DNA, and $9 \times 10^{-8}$ colonies per electroporated cell. Seventy-five percent (9/12) of PCR screened transformants contained the entire transgene construct, while the remaining contained truncated versions (Fig. 3b). Transgene integration was also achieved via biolistics, however we observed approximately an order of magnitude lower transformation efficiency relative to electroporation. Positive transformants showed a two- to fourfold increase in mCherry fluorescence over wild type (Fig. 3c), and confocal microscopy confirmed mCherry fluorescence localized to the nucleus and cytoplasm in these cells (Fig. 3d). Additional promoter configurations, with and without their respective introns, were also evaluated, including elongation factor 1-alpha 2 (*eef1A2*) and photosystem I reaction center subunit II (*psaD*), both utilizing the *eef1A2* terminator, which displayed comparable transformation efficiencies and mCherry fluorescence (Supplementary Data File 2). *P. renovo* is also sensitive to G418, which we have successfully used as a selection agent.

Figure 4a depicts the construct utilized for targeted engineering of the *P. renovo* chloroplast via biolistics. The native 16S ribosomal RNA promoter and 3′ UTR were utilized to direct transgene expression. The commonly utilized antibiotics spectinomycin and streptomycin failed to inhibit *P. renovo* growth, and the above utilized phleomycin (for nuclear transformation) was ineffective for isolation of viable chloroplast transformants. Therefore, we chose erythromycin for selection, following antibiotic sensitivity screening. Notably, there is 100% homology between the last 9 base pairs (anti-Shine-Dalgarno sequence) of the *P. renovo* 16S rRNA and the *E. coli* 16S rRNA[26]. Thus, a canonical *E. coli* ribosomal binding site (RBS, AGGAGGTTATAAAAA) was used to direct translation. The

erythromycin resistance gene (*ereB*) was linked to the reporter super folder green fluorescent protein (*sfGFP*) in an operon[27] for rapid identification of transgenic lines. When fully constructed with targeting homology arms, this plasmid readily yielded transformed microalgae using a conventional biolistic approach[28–30].

Transformants could be rapidly identified via the reporter gene by imaging of the bombarded plate in a gel imaging station with filter sets suitable for sfGFP detection. This procedure yielded an average ($n = 3$) of a single colony per transformation with efficiencies of 1.4 colonies per μg delivered DNA and $8 \times 10^{-9}$ colonies per microalgal cell. Colonies positive for sfGFP were passaged on selective media and proper integration of the construct into the target region was verified via PCR using primers binding outside the homology region and within the transgene operon, depicted in Fig. 4a, b. 38–48-fold greater sfGFP fluorescence was observed over wild type when measured via fluorometry (Fig. 3c). Epifluorescent microscopy showed sfGFP fluorescence successfully localized to the chloroplast (Fig. 4d).

## Discussion

*P. renovo* displayed a distinct rapid growth rate phenotype, in initial screening trials comparing over 100 unique isolates (Fig. 1). Peak growth rate at 35 °C (Supplementary Fig. 1b) and cultivation capacity up to ~3× seawater salinity (Supplementary Fig. 1d) indicates this strain is well-suited for outdoor cultivation in high temperature regions with saltwater access. These traits complement those of previously identified winter candidate deployment strains[15,31], laying the foundation for crop rotation strategies[17,32]. Given the potential discordance between optical density and biomass density we further characterized *P. renovo*'s biomass productivity; the diel biomass productivity reported here exceeds the target productivity of $25\,\mathrm{g\,m^{-2}\,day^{-1}}$ reported by Davis, et al.[33] for cost-competitive algal biofuels. Higher biomass

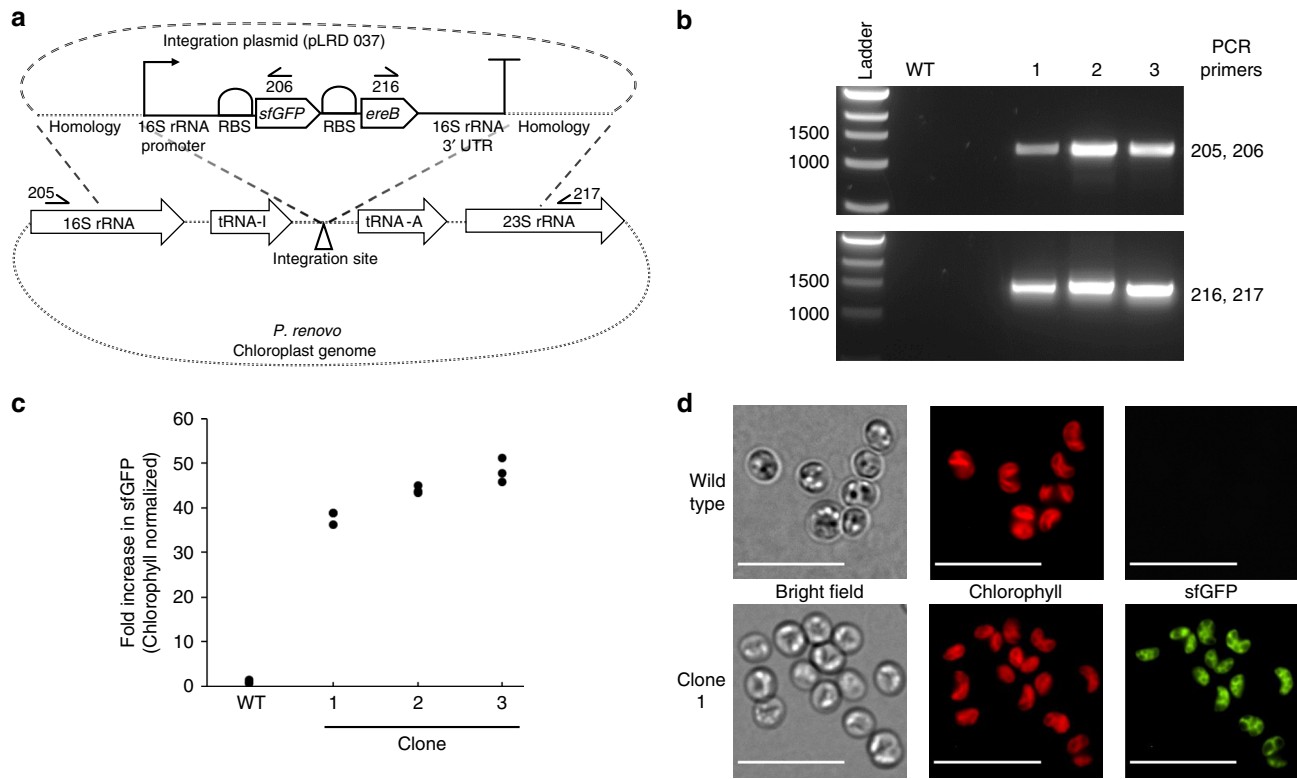

**Fig. 4** Overview of *P. renovo* chloroplast transformation. **a** Construct design showing genetic elements utilized and homology directed integration into the chloroplast genome, along with the primers used for subsequent PCR confirmation of transformants. **b** PCR verification of 3 clones utilizing primers shown in panel **a**. **c** Dot plot of fluorescent plate reader data of wild type and sfGFP transformants, normalized to chlorophyll autofluorescence. Data is from three biological replicates. **d** Epifluorescent microscopy images of wild type and sfGFP transformant microalgae. Red coloring represents chlorophyll autofluorescence, green coloring represents sfGFP fluorescence, 10 μm scale bar

productivities are likely achievable, given the suboptimal growth temperatures used in this study, which simulated outdoor cultivation in Mesa, Arizona. Future studies will evaluate outdoor productivity metrics to assess translatability of indoor metrics to outdoor systems in geographic regions better suited for high temperature cultivation.

Biomass analysis indicates the primary biomass hydrolysis product in *P. renovo* is glucose, which is a favorable feedstock for downstream biotechnological applications[34]. A drastic depletion of glucose was observed following inoculation into fresh media, similar to outdoor cultivation trends observed in other microalgal genera[15] (Fig. 2c). Dark period biomass loss is characterized almost exclusively by a decrease in glucose. These phenomena putatively function as a mechanism to remobilize glucose as an energy source for cell division and cellular homeostasis[35]. Thus, as previously reported[36], dark period biomass loss is an important parameter to consider when cultivating microalgae for biomass production. Under the conditions tested here, dark period biomass loss ranged from 0.25 to 0.46 g m$^{-2}$ h$^{-1}$. Combined, these data highlight the potential advantage of harvesting *P. renovo* biomass prior to dark period losses to maximize biomass and storage carbon yields.

Interestingly, cell division occurs during both the light and dark periods when grown under a diel cycle (Fig. 2a). This is contrary to many microalgae that synchronize cell division to occur at night; such is the case for the genera *Chlamydomonas*[37], *Nannochloropsis*[38], *Chlorella*, and *Scenedesmus*[39]. This continuous diurnal and nocturnal cell division, coupled with the compact genome(s), may partially explain the rapid doubling time and high biomass productivity of *P. renovo*, and represents

an area warranting further research. Cell division and biomass accumulation cease concurrently, suggesting a non-photosynthetic state when nitrogen-deprived (Fig. 2a). This is notable as some microalgae will continue to accumulate biomass post-nitrogen deprivation, presenting another avenue for comparative analyses[40]. Importantly, addition of nitrogen following growth arrest resulted in reinitiated growth, implicating nitrogen deprivation as the key driver for entry into stationary phase under the conditions evaluated in this study (Supplementary Fig. 1a). Optimization of nitrogen levels and harvest point may lead to enhanced productivity and storage carbon content.

Comparative transcriptomic analyses identified a series of previously unreported, halo-responsive genes. *dmc1*, which is involved in homologous chromosome pairing during meiosis was one of the most highly upregulated transcripts at higher salinity[41]. *rad54*, encoding a putative DMC1-interacting protein known to function during homologous recombination, is concurrently upregulated[42,43]. The upregulation of these genes could be attributed to meiosis, or homologous recombination repair of double strand DNA breaks, due to increased double strand breaks at higher salinities[44,45]. The observation of differentially expressed genes putatively associated with meiosis and homologous recombination suggests *P. renovo* may participate in sexual mating, and is capable of DNA repair via nuclear homologous recombination, both powerful tools for genetic manipulation. Indeed, sexual mating has been leveraged for trait stacking in both microalgae and plants and presents a powerful approach for rapid development of production hosts. Though we acknowledge that gene homology is insufficient evidence to assert functionality, as reviewed by Fučíková et al., multiple morphological/cytological

observations of syngamy and/or meiosis have been reported in the class Trebouxiophyceae and high conservation of meiotic genes is found within this class[46–48].

Downregulation of genes encoding proteins putatively relating to lipid remodeling was observed under high-salt conditions, including *pks1, pks15, ppsA, ppsC, iput1,* and *cerk. ppsA, ppsC, pks1,* and *pks15* are involved in the synthesis of diverse lipids and polyketides which have been implicated in cell wall permeability[49]. *cerk* is an enzyme that transfers a phosphate group to ceramide and is potentially acting in coordination with *iput*1 which transfers a glucuronic acid moiety to glycosyl inositol phosphorylceramides. Ceramides provide the lipid backbone for plant sphingolipids, and are primarily believed to be structural components of cellular membranes; however, ceramides have also been suggested to play a role in plant signaling[50]. The above data suggests that *P. renovo* is potentially using lipid remodeling to tune membrane permeability at differing salinities.

To facilitate *P. renovo* genetic and metabolic engineering, we developed tools enabling transgene expression in both the nucleus and chloroplast. Interestingly, only 9 of the 12 nuclear transgenic isolates screened showed insertion of the full transgene construct. Of the remaining three isolates, two were shown to have a truncated promoter or terminator, and one was shown to have an incomplete *mCherry* coding sequence, observed by the inability to generate a full-length coding sequence PCR product (Fig. 3b). It is not clear whether these truncated transgene constructs are the result of native *P. renovo* machinery cleaving the transgene construct or an incomplete PCR product integrating into the genome. Fluorescent plate reader analysis of the clones revealed increases in mCherry fluorescence over wild type for the 11 clones containing a full length *mCherry* coding sequence (Fig. 3b, c). As expected, the single clone without a full length *mCherry* coding sequence did not show an increase in mCherry fluorescence relative to wild type. The variation in mCherry fluorescence could be due to unique integration sites, or multiple integration events. mCherry fluorescence was primarily localized to the nucleus and cytoplasm of the transformant, with no observable chloroplast localization, as has been previously observed[51]. Preliminary analyses indicate stable nuclear transformation; mCherry fluorescence intensity remains constant following passaging on and off the selection marker.

Successful chloroplast transformation was phenotypically observed via high reporter expression, and epifluorescent microscopy confirmed successful localization of the sfGFP to the *P. renovo* chloroplast, evident by overlap with chlorophyll autofluorescence (Fig. 4c, d). The ability to confirm transgenic colonies via direct imaging of the high sfGFP signal will increase the throughput of control element screening, such as varied promoter strengths, in order to optimize metabolic engineering strategies. Additionally, the successful utilization of an *E. coli* RBS for operonic expression presents the potential for optimization of mRNA translation via the employ of established RBS prediction software[26]. Thus, these tools will be useful for biotechnological applications, such as overexpression of desired industrial enzymes[28,52] or fine-tuned regulation of native and/or synthetic metabolic pathways for bioproduct formation[53].

The transformation procedure presented herein is a facile protocol with relatively rapid turnaround time that can be completed in a few hours. Given the fast growth of this alga, transformant colonies can be generated in as few as 5 days, considerably faster than top-candidate deployment strains such as *Nannochloropsis*, wherein colonies need ~21 days of growth before verification analyses can be performed[7]. We have also provided the sequences of two additional nuclear promoters (elongation factor 1-alpha 2 and photosystem I reaction center subunit II, Supplementary Data File 2), which we have

successfully utilized to generate transformants. These additional promoters could prove useful for expression of multiple transgenes from one nuclear targeting cassette.

The full biotechnological potential of microalgae has yet to be brought to bear at commercial scale, in part due to the lack of robust, high-productivity strains suitable for outdoor deployment. Further, microalgal genetics in non-model systems has proven to be a limiting factor in strain development and fundamental mechanistic probing of top-candidate deployment strains. Here, we characterize a novel high-productivity, halophilic, thermotolerant microalga, and report the development of genomic and genetic tools therein. In addition to possessing a series of traits suitable for outdoor deployment, this strain displays favorable characteristics for development as a model system, in part due to its compact genomic architecture and rapid genetic throughput. Combined, the above-described traits present a unique complement to extant model systems. Further genetic development of *P. renovo* will enable both fundamental and applied insights, including elucidation of key regulatory mechanisms governing rapid growth and halotolerance in microalgae, as well as strain engineering strategies targeting enhanced productivity and carbon partitioning in a deployment-relevant microalga.

## Methods

**Strain screening and characterization of algal growth.** Microalgae were screened as previously reported[15], under conditions representative of summer cultivation. Briefly, 100 mL microalgal cultures were sparged with 2% $CO_2$ at 100 mL min$^{-1}$. Temperature cycled from 21 to 32 °C while lighting cycled from 0 to 965 µmol m$^{-2}$ s$^{-1}$ (the maximum output of the utilized lights). This regime was designed to simulate the temperature and lighting diel cycles measured in outdoor raceway ponds located at the Arizona Center for Algae Technology and Innovation testbed site located in Mesa Arizona, during the time frame of 12 June to 21 July 2014. We utilized a modified f/2 medium for cultivation, termed NREL Minimal Medium (NM2), in seawater (Gulf of Maine, Bigelow Laboratory), the following were added to the indicated final concentrations followed by addition of 12 M HCl to attain pH 8.0: NH$_4$Cl (5.0 × 10$^{-3}$ M), NaH$_2$PO$_4$·H$_2$O (0.313 × 10$^{-3}$ M), Na$_2$SiO$_3$·9H2O (1.06 × 10$^{-4}$ M), FeCl$_3$·6H$_2$O (1.17 × 10$^{-5}$ M), Na$_2$EDTA·2H$_2$O (1.17 × 10$^{-5}$ M), CuSO$_4$·5H$_2$O (3.93 × 10$^{-8}$ M), Na$_2$MoO$_4$·2H$_2$O (2.60 × 10$^{-8}$ M), ZnSO$_4$·7H$_2$O (7.65 × 10$^{-8}$ M), CoCl$_2$·6H$_2$O (4.20 × 10$^{-8}$ M), MnCl$_2$·4H$_2$O, (9.10 × 10$^{-7}$ M), thiamine HCl (2.96 × 10$^{-7}$ M), biotin (2.05 × 10$^{-9}$ M), cyanocobalamin (3.69 × 10$^{-10}$ M), Tris base (24.76 × 10$^{-3}$ M). For genetic engineering, the concentration of seawater was diluted 4-fold with Milli-Q water (Millipore Corporation), ammonium bicarbonate was utilized in the place of ammonium chloride, and 1.5× vitamins (thiamine HCl, biotin, cyanocobalamin) were utilized. Agar (Bacto) plates were prepared by autoclaving 3% agar in Milli-Q water, followed by addition of an equal volume of sterile filtered NM2 (seawater diluted twofold) with 2× nutrients, trace metals, vitamins, and Tris buffer. Sterile filtered selection antibiotic was added as necessary to appropriate concentrations, defined below.

To obtain a more detailed analysis of *P. renovo* growth, the above conditions were utilized with a 120 mL culture volume. Mid-log phase seed culture was generated under the above diel conditions, and used to inoculate 36, 120 mL cultures at a starting optical density of 1.0, in biological triplicate. Inoculation occurred approximately halfway through the light cycle, and biomass samples were harvested at the initiation, mid-point, and end-point of the lighting cycle, as indicated in Fig. 2. Sterile water was added prior to samplings to account for evaporative loses. Cell counts were performed using an Improved Neubauer hemocytometer. To convert volumetric productivities to areal values, the cross-sectional area of the culture tubes (0.00459 m$^2$) was employed.

Growth at varying salinities for Supplementary Fig. 1d were done in the same fashion as culture collection screening except salt levels were varied by addition of sea salts (Sigma S9883). 17.5 g L$^{-1}$ salinity was achieved via addition of seawater to milli-Q water and higher salinities utilized addition of sea salts. One hundred milliliters of culture was harvested after 6 days of growth, utilizing the temperature and light cycling from the culture collection screening methods described above. Temperature optima data, represented in Supplementary Fig. 1b was generated by growing strains in NM2, with culture conditions of constant 400 µmol m$^{-2}$ s$^{-1}$ lighting, 2% constant $CO_2$ sparging, and 100 mL volume. To determine growth rates, optical density (750 nm) measurements were taken daily, Supplementary Fig. 1b.

**Compositional analysis.** Compositional analysis was carried out as reported previously[15], with the following modification; a Carbopac PA1 HPAEC column was utilized for sugar monomer (carbohydrate) analysis. Protein was quantified via CHN (carbon, hydrogen, and nitrogen) analysis, utilizing an Elementar VarioEL cube CHN analyzer according to the manufacture's specifications. Briefly, a 5 mg

sample is combusted at 950 °C, and subsequent gasses are carried via helium to reduction and adsorption tubes utilizing an intake pressure of 1200 psi and ultimately detected with a thermal conductivity detector. A nitrogen-to-protein conversion factor of 4.78 was used[54].

**Genome sequencing, assembly, and annotation**. High molecular weight algal genomic DNA was extracted from cells imbedded in agarose, purified and concentrated using AMPure PB beads. The DNA was then fragmented using Covaris g-Tubes. Fragmented and purified DNA was processed for 20 kb SMRT bell library prep. The long insert libraries were size selected using a Blue Pippin instrument (Sage Sciences, Beverly, MA). The sequencing primer was annealed to the selected SMRT bell templates. The libraries were bound to DNA polymerase and loaded on the PacBio RSII for sequencing. Sequencing was completed using either C2/P4 or C3/P5 chemistry and 3-h movies. 8 SMRT cells of sequencing data were assembled with FALCON, version 0.2.2[55]. The final assembly includes 29 contigs with an assembled genome size of 14.4 Mbp. Estimated fold coverage of the PacBio reads was 270×.

Genome annotation was performed using the BRAKER (v2) training and annotation pipeline[56] utilizing the six sets of transcriptomic reads (described below in transcriptome response to salinity) to inform AUGUSTUS gene models[57,58]. Functional annotation of the 8902 genes was performed by InterProScan 5[59] and BLASTp searches against the UniProt[60] protein blast database; reported E-values reflect this methodology. The *P. renovo* genome assembly and annotation is available for download at the Greenhouse Knowledgebase (greenhouse.lanl.gov).

**Transcriptome response to salinity**. In order to identify genes putatively conferring halotolerance, cells were cultivated under low- and high-salinity conditions, corresponding to 8.75 g L⁻¹ and 35 g L⁻¹ sea salts. Cells grow at approximately the same growth rate under these conditions (Supplementary Fig. 1c). Cells adapted to the appropriate salinity level were grown in NM2 medium, utilizing ammonium bicarbonate as a nitrogen source. Biological triplicate culture conditions were as follows: 33 °C, 400 µmol m⁻² s⁻¹ lighting, and 2% constant $CO_2$ sparging in 100 mL volume. Seawater was employed as a source of salt, as this provides a more accurate proxy for halo-responsiveness compared to NaCl[61]. Seawater was diluted with distilled water to obtain appropriate salinity levels. The data from these methods are reflected in Supplementary Fig. 1c and salinity transcriptomics data. The above methods were done with the explicit goal of reducing culture shock, and subsequent global stress response, thus allowing a steady state comparison of RNA transcripts relating to salinity tolerance. RNA was obtained utilizing a QIAGEN RNeasy Plant Mini Kit following the manufacturer's recommendations, cells were homogenized under liquid nitrogen using a mortar and pestle. Paired-end 150 bp Illumina read RNA seq data were received from Genewiz in the form of compressed fastq files. Samples were comprised of two conditions and three biological replicates of each condition, resulting in six total samples. Raw fastq reads were quality trimmed using HTStream[62] and mapped via *Salmon*[63] to the available genomic assembly. Coding regions were extracted from the full reference assembly prior to mapping. Read counts were formatted into a tab-separated file and migrated to R[64] to perform differential expression using the edgeR[65] package. Low-level transcripts were filtered and removed, and all libraries were normalized to each other. Transcript counts were fit to a generalized linear model and the Cox-Reid profile-adjusted likelihood method was used to estimate the dispersion of each transcript. Differential expression was performed by a quasi-likelihood test between each condition. Transcripts were determined as differentially expressed when the corrected p-value (also known as q-value, or False Discovery Rate) was less than or equal to 0.05 after a Benjamini-Hochberg correction for multiple hypothesis testing.

Whole genome alignments to other publicly available *Picochlorum* genomes were done as follows: six assemblies of different strains of *Picochlorum sp.* were compared using the *nucmer* utility in the large-scale alignment program MUMmer[12,13,66,67]. Maximal matches were found and total bases matching between the samples were summed and the percent identity was reported as the average identity among the maximal unique matches.

Gene ontology analysis was performed as follows: differentially expressed genes were assigned putative functions by extracting the FASTA sequence from the original list of genes and aligning the sequence against the available *Chlamydomonas reinhardtii* annotated assembly (version 5.5) via BLAST[68]. Protein identification numbers and putative annotations were then uploaded to the UniProt[60] database and cross-referenced against the available gene ontology (GO) terms. GO terms were visualized on a semantic space scatterplot with the online software Revigo[69].

**Nuclear engineering**. A nuclear integration cassette, as depicted in Fig. 3a, was synthesized and subcloned into the pUC19 plasmid backbone by Genewiz, Inc (South Plainfield, NJ). The selection marker, 2A peptide and *mCherry* were codon optimized to the *P. renovo* genome. The final linear PCR product (from primers LRD 49 and 11) for transformation was generated utilizing Q5 2× hot start master mix (NEB) and purified with a PureLink Quick PCR Purification kit (Invitrogen) following the manufacture's protocol, modified to include a second wash step.

10 OD units (~475 × 10⁶ cells) of early log phase cells per transformation were harvested and washed three times at room temperature in 375 mM D-Sorbitol (Sigma S6021). Washing utilized 2 mL Eppendorf tubes, 950 µL of 375 mM D-Sorbitol per wash, centrifuged at 8000 × g for 1 min. After washing, cells were resuspended in 100 µL of 375 mM D-Sorbitol; 3 µg of DNA at 850 ng per µL (concentrated on a vacuum centrifuge) was added to the cells and gently mixed. Cells and DNA were incubated for 3 min, transferred to an ice cold 2 mm gap electroporation cuvette (Bulldog Bio) and electroporated with a Gene Pulser Xcell (Bio-Rad) electroporator utilizing a set time constant and voltage protocol of 2200 volts with a 25 ms time constant. Immediately following the pulse, cells were transferred to 400 µL of media supernatant (from the above utilized cells) and incubated at room temperature for 15 min. Cells were then split equally between three selection plates (1.5% agarose) comprised of NM2 supplemented with 20 µg per mL of phleomycin (InvivoGen). Plates were placed in a Percival incubator with fluorescent lighting at 33 °C, 150 µmol m⁻² s⁻¹, and 1.5% $CO_2$. Colonies were counted and passaged on selection after 5 days for further analysis. A table of all DNA fragments and PCR primers utilized in this study is supplied in the supplementary information (Supplementary Data File 2).

**Chloroplast engineering**. Homology arm sequences were PCR amplified from chloroplast genomic DNA using NEB Q5 Master Mix from New England Biolabs (Ipswich, MA). A promoter-RBS-*sfGFP*-RBS-*ereB*-3′ UTR cassette was synthesized by Genewiz, Inc (South Plainfield, NJ) as depicted Fig. 4a. The chloroplast integration cassette was assembled into a pUC19 backbone using 2× Gibson Assembly Mix from New England Biolabs, following the manufacturer's protocol. Complete vector sequences were confirmed by Sanger sequence analysis (Genewiz, South Plainfield, NJ).

Biolistic transformation was employed to deliver DNA into the chloroplast, as reported previously[28,70]. Ten micrograms of plasmid DNA (QIAprep spin miniprep kit QIAGEN) was precipitated onto 550 nm gold sphere nanoparticles (Seashell Inc.) under constant vortexing. Ten microliters of plasmid DNA (1 µg/µL) was added to 60 µL of gold particles (50 mg per mL), followed by dropwise addition of 50 µL of 2.5 M $CaCl_2$ and 20 µL of 0.1 M spermidine (Sigma S0266-1G). This was vortexed for 5 min, incubated for 1 min at room temperature, briefly centrifuged, and washed with 140 µL of isopropanol. Following removal of wash supernatant, the gold particles were resuspended in 60 µL of isopropanol and gently sonicated in a bath sonicator to resuspend the pellet. To assay loading efficiency of the DNA onto the gold, a 9 µL aliquot was taken, washed in 9 µL of water and assayed for DNA concentration utilizing a NanoDrop 2000 spectrophotometer.

To transform *P. renovo*, an overnight culture was grown to early log phase in NM2, concentrated to 2.5 OD units in 170 µL, and spread evenly onto a 100 × 15 mm NM2 agar plate supplemented with 800 µg per mL erythromycin (Sigma E5389-5G). A Biolistic PDS-1000/He Particle Delivery System (metal case version) (Bio-Rad) was used for bombardment, which was accomplished by drying 9 µL of the above DNA loaded gold particles onto the macrocarrier (fast, low humidity drying was accomplished by placing the loaded macrocarrier into the bombardment chamber and pulling vacuum[71]), and bombarding cells 5 cm below the macrocarrier with a 1100 psi rupture disk. After bombardment, plates were placed into the same growth conditions described above. Following 7 days of growth, the plates were imaged with a FluorChemQ gel imaging station (Protein Simple) with 475/35 and 573/35 nm respective excitation and emission filters, which allowed direct imaging of sfGFP positive colonies.

To assess construct integration into the genomes of *P. renovo*, genomic DNA was extracted from cells passaged on the selection marker utilizing a MasterPure™ Yeast DNA Purification kit (Lucigen). PCR was performed utilizing Q5 Hot Start High-Fidelity polymerase (New England Biolabs) according to the manufacturer's recommendations. A table of the utilized primers is provided in the supplementary information (Supplementary Data File 2). Images of uncropped gels (depicted in Figs. 3 and 4) are provided in Supplementary Fig. 5.

**Fluorescent plate reader analysis**. Colonies were restreaked onto fresh agar plates supplemented with the appropriate selection marker (phleomycin 20 µg per mL and erythromycin 800 µg per mL for the nucleus and chloroplast, respectively) and grown in triplicate in 3 mL of growth media (no selection marker) in standard glass cell culture tubes mixed daily via vortexing. Cultures were grown in the above described Percival incubator conditions (33 °C, 150 µmol m⁻² s⁻¹, 1.5% $CO_2$). Early log phase cells were analyzed for mCherry and sfGFP fluorescence utilizing 200 µL of cell culture in a black 96 well plate and a FLUOstar Omega plate reader v. 5.11 R3 (BMG Labtech). To quantify mCherry a 584 nm excitation filter and 620/10 nm emission filter were utilized with gain set to 2500; to quantify sfGFP a 485/12 nm excitation and 520 nm emission filter set was used with gain set to 1200. Data was normalized to chlorophyll content, which was determined by using a 485/12 nm excitation and 680/10 emission filters, with gain set to 1500. Data is represented as a fold increase over the wild type alga.

**Statistics and reproducibility**. All experiments in this study utilized triplicate biological replicates. Error is represented by the sample standard deviation of the

replicates, unless otherwise noted. All source data for main text figures and charts is provided in Supplementary Data File 3.

**Microscopy**. Mid-log phase chloroplast sfGFP transformants and wild type were imaged with a Nikon Eclipse 80i microscope, equipped with a Nikon Intensilight C-HGFI mercury lamp light source, a Nikon Plan Apo VC 100× objective lens, and a Nikon DS-QiMc camera. NIS-Elements BR 4.30.01 software was utilized for imaging chlorophyll and sfGFP (31017 – Chlorophyll Bandpass Emission and 41017 – Endow GFP/EGFP Bandpass, both from CHORMA®). Imaging of wild type and transgenic lines employed equivalent exposure time and gain settings. ImageJ was used for post imaging analysis.

Nuclear mCherry transformants and wild type were imaged with a Nikon C1si confocal microscope, equipped with EZ-C1 3.60 software. Chlorophyll was imaged with a 650 LP filter. mCherry was imaged with a 590/50 filter. Both chlorophyll and mCherry were excited with a 561.4 nm laser. Laser intensity, pin hole size, pixel dwell time, and gain were set using an mCherry clone. Equivalent settings were utilized for imaging wild type cells. ImageJ was used for post imaging analysis.

**Reporting summary**. Further information on research design is available in the Nature Research Reporting Summary linked to this article.

## Data availability

The algal strain (*Picochlorum renovo*), DNA elements, and raw data supporting the conclusions of this manuscript will be made available by the authors, without undue reservation, to any qualified researcher. The genome for *Picochlorum renovo* is publicly available at https://greenhouse.lanl.gov/greenhouse/organisms. Genomic sequence data can also be accessed at NCBI, Project Number PRJNA558990. The raw data supporting the conclusions of salinity transcriptomics is available from the Sequence Read Archive, under Project Number PRJNA553204.

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

## Acknowledgements

The authors would like to thank Dr. Brian Vogler for insightful discussions regarding design and delivery of nuclear DNA constructs into eukaryotic microalgae, and Andy Politis, Bonnie Panczak, and Brittany Thornton for assistance in compositional analyses. This research was supported by the Department of Energy, Office of Energy Efficiency and Renewable Energy (EERE) under Agreements No. 22000 and DE-NL0029949. The views and opinions of the authors expressed herein do not necessarily state or reflect those of the United States Government or any agency thereof. Neither the United States Government nor any agency thereof, nor any of their employees, makes any warranty, expressed or implied, or assumes any legal liability or responsibility for the accuracy, completeness, or usefulness of any information, apparatus, product, or process disclosed, or represents that its use would not infringe privately owned rights.

## Author contributions

L.D., M.P., and M.G. designed and carried out initial strain screening to identify *P. renovo*. L.D. designed and carried out detailed characterization of *P. renovo* growth, S.V. performed compositional analysis. Genome sequencing and annotation was done by Y.K., B.H., S.S., and A.G. Salinity transcriptomics were designed and evaluated by L.D. and M.G., A.G. performed gene ontology and differential expression analysis, along with whole genome alignments. Nuclear and chloroplast constructs were designed and evaluated by L.D., C.H., J.L., M.P. and M.G.

## Competing interests

The authors declare no competing interests.
