## [Peer Review File (redacted) · Communications Biology]

Reviewers' comments:

Reviewer #1 (Remarks to the Author):

This article characterizes a species of microalgae with interesting properties. They claimed to do genomics, transcriptomics, and develop transgenic tools. These are new developments for this organism, and should be of interest to others in the field. On a more subjective note, the paper should influence thinking in the field.

I have one important notice [1] and a few suggestions for minor revisions [2-6]:

[1]: I did not see a way to access the data. The genome and annotations should be supplemental datasets [Dataset S1,S2, etc], but currently are only referenced at Line 410. It says they are at greenhouse.lanl.gov but when I looked, it was not there. Thus, data availability and accessibility should be enhanced. I also suggest uploading to NCBI and Mendeley (or another file hosting service with a DOI).

[2]: The subject, focus, and coverage of the study are great. In terms of a literary perspective, I would like to see the article written in a more active, succinct voice.

[3]: Lines 155-157: the source for the other *Picochlorum* genomes should be referenced.

[4]: In lines 180-185, the E-values, or other confidence measures should be reported for each gene.

[5]: In lines 241-242, glucose should not be counted as a starch

[6]: line 343: That microalgae were screened by sparging with CO₂ doesn't make sense. Perhaps this could be re-written. The authors could replace "microalgae were screened by sparging" with "microalgal cultures were sparged..."

Reviewer #2 (Remarks to the Author):

The manuscript "Development of a halophilic, thermotolerant model microalga, *Picochlorum renovo*" provides extensive characterization of the growth performance, genome sequence, and nuclear and chloroplastic transformation tools for the organism. It is the third genome reported for the genus, however neither of the other genome papers addressed genetic transformation potentials. On the whole, the manuscript is well written and well organized. Specific comments follow

1. The title suggests the authors are promoting this organism as a model for mechanistic studies of phototrophic organisms and yet the introduction seems to suggest *P. renovo* is noteworthy primarily due to its production phenotypes. Which is it? The context of the manuscript would benefit from revision of the introduction to address the characteristics most beneficial in a model system and which of those are present in *P. renovo*. Are the authors really suggesting it is on par with *Chlamydomonas*?

If not, perhaps suggest an argument for why additional "model" systems are needed and for what purpose? Alternatively, perhaps the title should be changed and the organism promoted as a candidate "production" strain with facile options for genetic improvement of its impressive growth phenotypes.

2. p. 4 line 65. *P. renovo* seems to be hot temperature strain rather than one for year-round cultivation, so the concept of crop rotation might be useful to provide context to the study and complement the early work by a subset of the authors on strains useful for cold temperature cultivation.

3. P. 5, lines 87-89. I believe these sentences and the refs belong at the very beginning of the results section.

4. Fig. 1. It is not obvious to me that *P. renovo* has a shorter lag phase than the other strains shown. That claim seems to be based on four time points and no error bars and would therefore require additional documentation.

5. p. 8, line 150. The authors' identification of potential meiosis genes would benefit from clarification of the value of meiotic recombination and reassortment in the context of the reflections suggested in item 1 above. Also, were all these genes found in the other *Picochlorum* genomes? This should be addressed as well.

6. Please clarify if linear DNA fragments prepared for nuclear transformation contained homology arms for targeting (it appears not). If so, then was targeted recombination attempted and unsuccessful? Were any insertion sites identified? Do the results presented represent transient or stable transformation? What data support the answer?

7. p. 11, line 202. "*P. renovo* is also sensitive to G418 WHICH (not and) can be successfully...."

Reviewer #3 (Remarks to the Author):

Overview:

In the manuscript "Development of a halophilic, thermotolerant model microalga, *Picochlorum renovo*" Dahlin et. al lay out a roadmap for selection through characterization and domestication of a potential industrially relevant microalgal strain. The authors nicely document biomass compositional changes for the strain throughout batch growth including FAME, protein and carbohydrates (with hydrolyzed monomers) to identify glucose (most likely in starch) as a rapidly utilized metabolite during diel growth. Ranges for culturing in varied salinity and temperature were determined. The strain was sequenced and annotated genomes were assembled for the nucleus, mitochondria and chloroplast. This allowed for salt treatment transcriptomics as well as initial tests showing proof of concept genetic engineering of both the chloroplast and the nucleus. Although genetic engineering in industrially relevant strains has been reported before (ie. Kilian 2011, Ajjawi 2017) Dahlin et. al show that characteristics of *Picochlorum renovo* would allow this chassis to be more suitable for non-biofuel related biotechnology aimed at outdoor cultivation based on environmental factor tolerance as well as the reduced turnaround time for transformation cycles. Below are questions/comments for the authors.

1. Reference #6 authored in part by M. Posewitz describes *Picochlorum celeri* as an extremely fast

growing algal strain. Was this strain also evaluated in initial screen used for selection of *P.renovo*? And if so, is that data shown in Figure 1?

2. Figure 1 shows an impressive OD growth curve for *P.renovo* which is the reason it was a clear winner. Was there any further characterization of the strain indicated by the grey line below. *Picochlorum* having a very small size would undoubtedly have a high OD/Biomass ratio. The strain indicated by the grey line may also have an impressive biomass productivity yet a lesser OD/biomass ratio, leading to a less striking visual on the graph.

3. In the section on genomic analysis there is discussion of several meiosis related genes as determined by automated annotation of the genome. References # 18-23 document that members of the class Chlorophyceae (*Volvox* and *Chlamydomonas*) undergo meiosis. Are the authors aware of any documented cases of members of class Trebuxiophyceae (or any non-Chlorophycean chlorophyte) meiosis? If not, it might be a stretch to infer function based on gene homology alone.

4. Was there a reason why bombardment was used for chloroplast transformation while electroporation was used for nuclear transformation? Additionally, was there logic to using mCherry vs GFP or could GFP have been used for both methods?

5. In the methods the light regime is "designed to simulate the conditions measured in outdoor raceway ponds located at the Arizona Center for Algae Technology and Innovation" yet the total output of the lights is 965 $\mu\text{mol}/\text{m}^2/\text{s}$. That's far less than the max seen at noon in Arizona summer months. Is it just the diel cycle timeline but not total photon flux what mimics the Arizona site?

Reviewer #1 (Remarks to the Author):

This article characterizes a species of microalgae with interesting properties. They claimed to do genomics, transcriptomics, and develop transgenic tools. These are new developments for this organism, and should be of interest to others in the field. On a more subjective note, the paper should influence thinking in the field.

I have one important notice [1] and a few suggestions for minor revisions [2-6]:

[1]: I did not see a way to access the data. The genome and annotations should be supplemental datasets [Dataset S1,S2, etc], but currently are only referenced at Line 410. It says they are at greenhouse.lanl.gov but when I looked, it was not there. Thus, data availability and accessibility should be enhanced. I also suggest uploading to NCBI and Mendeley (or another file hosting service with a DOI).

We apologize for the lack of data availability at the time of submission. The genome is now publicly accessible on the Greenhouse website (<https://greenhouse.lanl.gov/greenhouse/organisms/>). Further, we have uploaded all transcriptome sequence data to the NCBI Sequence Read Archive (SRA), Project Number PRJNA553204. Genome sequences have also been deposited at NCBI, Project Number PRJNA558990 (awaiting NCBI approval; accession number will be added to the manuscript upon receipt). These accession numbers have been added to the manuscript's Data Accessibility section (Line 594-601 in final untracked document). We have respectfully declined to append supplemental datasets, as we anticipate ongoing genome refinement; to ensure reader accessibility to current data, we will continue to update these public repositories with any newly generated versions of the genome. Pertinent text edits are as follows:

“...The genome for *Picochlorum renovo* is publicly available at <https://greenhouse.lanl.gov/greenhouse/organisms>. The raw data supporting the conclusions of salinity transcriptomics is available from the Sequence Read Archive, under Project Number PRJNA553204 (<https://www.ncbi.nlm.nih.gov/Traces/study/?acc=PRJNA553204>)...”

[2]: The subject, focus, and coverage of the study are great. In terms of a literary perspective, I would like to see the article written in a more active, succinct voice.

We thank the Reviewer for their complimentary assessment of the manuscript's subject, focus, and coverage. We have altered language throughout the manuscript to assume a more active, and where possible, succinct voice. Please refer to track changes in the accompanying revised file.

[3]: Lines 155-157: the source for the other *Picochlorum* genomes should be referenced.

We have updated the manuscript to reflect the source of all genome sequence data (please refer to Supplemental Table 1, Line 628) employed for comparative analyses.

[4]: In lines 180-185, the E-values, or other confidence measures should be reported for each gene.

We have updated the manuscript throughout to include E-values for all gene annotations. Please refer to Lines 147-156, and 184-189 of the updated manuscript (with track changes). Pertinent text edits are as follows:

“... The nuclear genome contains a homolog of the universally conserved meiosis associated gene, *spo11-2* (E-value: 2E-22), and homologs of multiple meiotic and/or homologous recombination associated genes (with reported E-values), including four *rad51* homologs (2E-107 to 4E-26), *dmc1* (2E-121), *pol2A* (0), *rfc1* (4E-34), *polD1* (0), *mre11* (2E-77), *rad50* (0), *rad54* (4E-134), *mus81* (6E-15), *msh4* (2E-10), *msh5* (2E-50), *rpa1* (3E-69), *rpa2* (8E-25), and *rpa3* (5E-16)¹⁸. Further evidence of genes putatively involved in meiosis is provided by the identification of *oda2* (0) and *bug22* (1E-72) homologs, which are flagella associated genes, implicated in gamete pairing prior to mating¹⁹⁻²³. We also identified a putative chlorophyllide-a oxygenase (0), necessary for chlorophyll b production, and cell division was observed to occur by autosporulation (Supplemental Figure 2)...”

“...A series of previously unreported haloresponsive genes were also observed, including *ppsA* (E-value: 2E-54), *ppsC* (E-value: 2E-60), *pks1* (E-value: 4E-30) and *pks15* (E-value: 1E-28) (polyketide synthases), *iput1* (inositol phosphorylceramide glucuronosyltransferase, E-value: 9E-75), *cerk* (ceramide kinase), *rad54* (DNA repair and recombination protein, E-value: 2E-20), and *dmc1* (disrupted meiotic cDNA 1, E-value 2E-121), discussed further below...”

[5]: In lines 241-242, glucose should not be counted as a starch

We have altered the language in the manuscript to more accurately reflect glucose as the primary monosaccharide product of biomass hydrolysis (Line 139-140, 253-254). Pertinent text edits are as follows:

“...Notably there is a substantial decrease in glucose (derived from biomass hydrolysis) following inoculation into fresh media, with glucose declining from 52% to 1.4% of AFDW (ash-free dry weight) (Figure 2c)...”

“...Biomass analysis indicates the primary biomass hydrolysis product in *P. renovo* is glucose, which is a favorable feedstock for downstream biotechnological applications³⁴...”

[6]: line 343: That microalgae were screened by sparging with CO2 doesn't make sense. Perhaps this could be re-written. The authors could replace “microalgae were screened by sparging” with “microalgal cultures were sparged...”

We thank the Reviewer for their suggestion and have altered the language in the manuscript to clarify this wording (Line 365). Pertinent text edits are as follows:

“...Microalgae were screened as previously reported ¹⁵, under conditions representative of summer cultivation. Briefly, 100 mL microalgal cultures were sparged with 2% CO₂ at 100 mL/min...”

Reviewer #2 (Remarks to the Author):

The manuscript "Development of a halophilic, thermotolerant model microalga, *Picochlorum renovo*" provides extensive characterization of the growth performance, genome sequence, and nuclear and chloroplastic transformation tools for the organism. It is the third genome reported for the genus, however neither of the other genome papers addressed genetic transformation potentials. On the whole, the manuscript is well written and well organized. Specific comments follow

1. The title suggests the authors are promoting this organism as a model for mechanistic studies of phototrophic organisms and yet the introduction seems to suggest *P. renovo* is noteworthy primarily due to its production phenotypes. Which is it? The context of the manuscript would benefit from revision of the introduction to address the characteristics most beneficial in a model system and which of those are present in *P. renovo*. Are the authors really suggesting it is on par with *Chlamydomonas*? If not, perhaps suggest an argument for why additional "model" systems are needed and for what purpose? Alternatively, perhaps the title should be changed and the organism promoted as a candidate "production" strain with facile options for genetic improvement of its impressive growth phenotypes.

We believe *P. renovo* represents both a promising production host and an emerging model for mechanistic studies. However, our primary intent is to present a promising production organism to the algal research community. Thus, we have followed the Reviewer's guidance and altered the title of the manuscript to more accurately reflect this intent, as follows: Development of a high-productivity, halophilic, thermotolerant microalga, *Picochlorum renovo*.

Given *P. renovo*'s favorable deployment traits and rapid genetics, it has the potential to complement extant model systems. However, at present, *P. renovo*'s current level of development is certainly not on par with *C. reinhardtii* and we recognize the continued exceptional value of *Chlamydomonas* for basic science pursuits and biotechnological applications. Thus, it is premature to assert *P. renovo* is an established model system, but it is our hope that this manuscript will lay the foundation for development as such, as indicated in the manuscript Conclusions. (Line 346-360).

2. p. 4 line 65. *P. renovo* seems to be hot temperature strain rather than one for year-round cultivation, so the concept of crop rotation might be useful to provide context to the study and complement the early work by a subset of the authors on strains useful for cold temperature cultivation.

The Reviewer is correct that *P. renovo* is best-suited for Summer (hot temperature) deployment and thus complements previously identified candidate cold-weather deployment strains. We have followed the Reviewer's guidance and altered the text to introduce the concept of crop rotation (Lines 243-244). Pertinent text edits are as follows:

“... These traits complement those of previously identified winter candidate deployment strains^{15,31}, laying the foundation for crop rotation strategies^{17,32}...”

3. P. 5, lines 87-89. I believe this sentences and the refs belong at the very beginning of the results section.

We have followed the Reviewer's guidance and altered the text accordingly (Lines 107-109). Pertinent text edits are as follows:

“...We screened a >300-strain algal culture collection, in order to identify halotolerant strains^{14,15}...”

4. Fig. 1. It is not obvious to me that *P. renovo* has a shorter lag phase than the other strains shown. That claim seems to be based on four time points and no error bars and would therefor require additional documentation.

We agree our data, as presented, does not clearly provide evidence for a shorter lag phase; thus, we have removed all references to a shorter lag phase from the text.

5. p. 8, line 150. The authors identification of potential meiosis genes would benefit from clarification of the value of meiotic recombination and reassortment in the context of the reflections suggested in item 1 above. Also, were all these genes found in the other *Picochlorum* genomes? This should be addressed as well.

We thank the Reviewer for this suggestion; we have added text to note the potential value of meiotic recombination for the rapid development of production strains with stacked traits (Lines 324-326). With regards to the presence of meiotic genes in other *Picochlorum* genomes, we note that meiotic genes are ubiquitous in all genera of the class Trebouxiophyceae, as reported by Fučíková, et al (<https://doi.org/10.1111/jpy.12293>) and are thus not unique to *P. renovo*. We have added and referenced these findings in the manuscript text, lines 289-294. Pertinent text edits are as follows:

“... Indeed, sexual mating has been leveraged for trait stacking in both microalgae and plants and presents a powerful approach for rapid development of production hosts. Though we acknowledge that gene homology is insufficient evidence to assert functionality, as reviewed by Fučíková et al., multiple morphological/cytological observations of syngamy and/or meiosis have been reported in the class Trebouxiophyceae and high conservation of meiotic genes is found within this class⁴⁶⁻⁴⁸...”

6. Please clarify if linear DNA fragments prepared for nuclear transformation contained homology arms for targeting (it appears not). If so, then was targeted recombination attempted and unsuccessful? Were any insertion sites identified? Do the results presented represent transient or stable transformation? What data support the answer?

The Reviewer is correct; we did not include homology arms for the nuclear engineering work. We have updated the text to explicitly reflect the randomly integrative nature of the utilized cassette (line 191). Targeted nuclear integration has not yet been pursued but will be

investigated in follow-on studies in parallel with insertion site identification. Preliminary analyses indicate stable nuclear transformation; mCherry fluorescence remains constant following passaging on and off the selection marker. We have updated the text to reflect these findings (Lines 322-324). Pertinent text edits are as follows:

“We randomly integrated a linear PCR amplicon comprised of native promoter and terminator elements into the nuclear genome of *P. renovo*, directing transcription of 2A peptide-linked bleomycin resistance gene and the fluorescent reporter mcherry (Figure 3a) via electroporation.”

“...Preliminary analyses indicate stable nuclear transformation; mCherry fluorescence intensity remains constant following passaging on and off the selection marker...”

7. p. 11, line 202. "P. renovo is also sensitive to G418 WHICH (not and) can be successfully...."

We have followed the Reviewer's guidance and altered the language in the manuscript to reflect this change (line 210). Pertinent text edits are as follows:

“...*P. renovo* is also sensitive to G418, which we have successfully used as a selection agent...”

Reviewer #3 (Remarks to the Author):

Overview:

In the manuscript “Development of a halophilic, thermotolerant model microalga, *Picochlorum renovo*” Dahlin et. al lay out a roadmap for selection through characterization and domestication of a potential industrially relevant microalgal strain. The authors nicely document biomass compositional changes for the strain throughout batch growth including FAME, protein and carbohydrates (with hydrolyzed monomers) to identify glucose (most likely in starch) as a rapidly utilized metabolite during diel growth. Ranges for culturing in varied salinity and temperature were determined. The strain was sequenced and annotated genomes were assembled for the nucleus, mitochondria and chloroplast. This allowed for salt treatment transcriptomics as well as initial tests showing proof of concept genetic engineering of both the chloroplast and the nucleus. Although genetic engineering in industrially relevant strains has been reported before (ie. Kilian 2011, Ajjawi 2017) Dahlin et. al show that characteristics of *Picochlorum renovo* would allow this chassis to be more suitable for non-biofuel related biotechnology aimed at outdoor cultivation based on environmental factor tolerance as well as the reduced turnaround time for transformation cycles. Below are questions/comments for the authors.

1. Reference #6 authored in part by M. Posewitz describes *Picochlorum celeri* as an extremely fast growing algal strain. Was this strain also evaluated in initial screen used for selection of *P. renovo*? And if so, is that data shown in Figure 1?

During initial screening we did not compare *P. renovo* to *P. celeri*, as the latter strain had not yet been identified. *P. celeri* is not currently available in a public repository, however we are in the process of procuring this strain (subject to a Materials Transfer Agreement) in order to conduct follow-on *Picochlorum spp.* interspecies comparative analyses.

2. Figure 1 shows an impressive OD growth curve for *P.renovo* which is the reason it was a clear winner. Was there any further characterization of the strain indicated by the grey line below. Picochlorum having a very small size would undoubtedly have a high OD/Biomass ratio. The strain indicated by the grey line may also have an impressive biomass productivity yet a lesser OD/biomass ratio, leading to a less striking visual on the graph.

Yes, we did characterize the strain indicated by the grey line in Figure 1. Ash-free dry weight analysis indicated this strain had a lower biomass productivity than *P. renovo*. With regards to OD/biomass ratios, we agree with the Reviewer's assessment; end-point ash-free dry weight was a primary criterion for screening and down-selection (as described in Dahlin, et al 2018; <https://doi.org/10.3389/fpls.2018.01513>) and indeed, led to the identification of a second promising deployment candidate, that while having a lower OD/biomass ratio, is also highly productive (not pictured in Figure 1). We are currently preparing a manuscript describing this strain. We have noted the potential discordance between OD and biomass in the text (Line 245-246). Pertinent text edits are as follows:

“...Given the potential discordance between optical density and biomass density we further characterized *P. renovo*'s biomass productivity; the diel biomass productivity reported here exceeds the target productivity of 25 g/m²/day reported by Davis, et al. ³³...”

3. In the section on genomic analysis there is discussion of several meiosis related genes as determined by automated annotation of the genome. References # 18-23 document that members of the class Chlorophyceae (*Volvox* and *Chlamydomonas*) undergo meiosis. Are the authors aware of any documented cases of members of class Trebuxiophyceae (or any non-Chlorophycean chlorophyte) meiosis? If not, it might be a stretch to infer function based on gene homology alone.

Fučíková, et al (<https://doi.org/10.1111/jpy.12293>) have reviewed several morphological/cytological observations of syngamy and/or meiosis in the class Trebuxiophyceae and further reported high conservation of meiotic genes within this class. However, we agree with the Reviewer that gene homology is insufficient evidence to infer functionality. We have modified the text to note these observations and the putative nature of meiosis in *P. renovo* based upon the presented lines of evidence (Lines 289-295). Pertinent text edits are as follows:

“Indeed, sexual mating has been leveraged for trait stacking in both microalgae and plants and presents a powerful approach for rapid development of production hosts. Though we acknowledge that gene homology is insufficient evidence to assert functionality, as reviewed by Fučíková et al., multiple morphological/cytological observations of syngamy and/or meiosis have been reported in the class Trebouxioophyceae and high conservation of meiotic genes is found within this class ⁴⁶⁻⁴⁸...”

4. Was there a reason why bombardment was used for chloroplast transformation while electroporation was used for nuclear transformation? Additionally, was there logic to using mCherry vs GFP or could GFP have been used for both methods?

Initial nuclear transformation trials utilized electroporation, as this had been previously demonstrated in a number of algae, including *Nannochloropsis spp.* Follow-on studies have also

successfully achieved transgene integration via biolistics, however approximately an order of magnitude lower transformation efficiencies were achieved relative to electroporation. With regards to chloroplast engineering, bombardment is routinely used for transformation of higher plant chloroplasts and the *Chlamydomonas* chloroplast, presumably due to the additional membrane(s) surrounding the chloroplast, making DNA entry more difficult. Thus, we hypothesized bombardment would yield the highest efficiencies for chloroplast transformation. To date, we have not achieved chloroplast transformation via electroporation. We have update the text to reflect this (lines 201-203).

“Transgene integration was also achieved via biolistics, however we observed approximately an order of magnitude lower transformation efficiency relative to electroporation (data not shown).”

With regards to fluorescent protein selection, initial trials for both nuclear and chloroplast transformation employed GFP. However, low copy number (presumably single copy) led to a low GFP signal in the nucleus. mCherry was thus selected as an alternative, based upon prior work by the Mayfield group showing mCherry has the highest signal to noise ratio in *Chlamydomonas* (doi: 10.1111/tpj.12165). Higher chloroplast expression (presumably due to higher copy number) was sufficient for plastidial GFP detection, and thus no alternative fluorescent protein was evaluated. We have altered the text to reflect the basis for mCherry selection (Line 194-195). Pertinent text edits are as follows:

“mCherry was chosen as a reporter gene for nuclear expression based on prior reports of high signal to noise ratios in microalgae ²⁵...”

5. In the methods the light regime is “designed to simulate the conditions measured in outdoor raceway ponds located at the Arizona Center for Algae Technology and Innovation” yet the total output of the lights is 965 $\mu\text{mol}/\text{m}^2/\text{s}$. That’s far less than the max seen at noon in Arizona summer months. Is it just the diel cycle timeline but not total photon flux what mimics the Arizona site?

The Reviewer is correct, the LED lights utilized for screening are below the maximum incident light observed outdoors in Arizona. Thus, our outdoor simulation explicitly mimics the temperature and diel *cycle* of lighting. We have altered the text to clarify simulation parameters (Lines 367-370), as follows:

This regime was designed to simulate the temperature and lighting diel cycles measured in outdoor raceway ponds located at the Arizona Center for Algae Technology and Innovation testbed site located in Mesa Arizona, during the time frame of June 12th to July 21st, 2014.

REVIEWERS' COMMENTS:

Reviewer #1 (Remarks to the Author):

The manuscript looks great, I have no further revisions to suggest.

I suggest acceptance and publication.

Reviewer #2 (Remarks to the Author):

The authors have satisfactorily answered or addressed my concerns. My opinion is that the manuscript is ready for publication.

Reviewer #3 (Remarks to the Author):

The authors of "Development of a high-productivity, halophilic, thermotolerant model microalga, *Picochlorum renovum*." have revised the manuscript with updates and clarifications to all my questions and concerns. At this point I would recommend this work for publication.